# BEYOND MOMENT : RETHINKING EVALUATION PARADIGM FOR TIMELINE SUMMARIZATION IN THE ERA OF LLMS

## ABSTRACT

Timeline summarization (TLS) aims to condense large collections of temporally ordered documents into concise and coherent narratives of key events. While recent advances with large language models (LLMs) have improved, progress in TLS cannot be assessed objectively due to the lack of reliable evaluation metrics. Existing evaluation metrics rely on the assumption that milestones aligned at the same timestamp convey identical semantic meaning. This design choice inherently biases against abstractive or semantically equivalent outputs while emphasizing temporal consistency (Date F1 and A-ROUGE). Consequently, such evaluation protocols fail to adequately reflect the genuine improvements brought by LLMs and deviate from human judgments when comparing the relative merits of different methods. To more faithfully assess whether the predicted timeline and the reference timeline truly refer to the same events, we propose a new evaluation framework in which all metrics are grounded on semantically aligned sentence pairs rather than merely time-aligned milestones. We leverage LLMs to compute semantic similarity, align sentence pairs via maximum-weight bipartite matching, and compute the Sematic-Alignment Score. Building on this alignment, Date-F1 and ROUGE metrics are further introduced to jointly evaluate semantic coverage and temporal fidelity, which we term Semantic-Alignment Date-F1 and Sematic-Alignment-ROUGE, respectively. To validate the effectiveness of our proposed metrics, we introduce a Full-Stage LLM-TLS approach and conduct comparisons against prior methods. Experiments demonstrate that FS-LLM-TLS not only surpasses prior methods on existing evaluation metrics but also that its advantages are more faithfully and effectively reflected under our evaluation framework, which offers a more comprehensive assessment of method quality. This evaluation framework establishes a new paradigm for TLS evaluation, laying the foundation for future experimentation and system development. [1] [2]

## 1 INTRODUCTION

Timeline summarization (TLS) condenses large collections of temporally ordered documents into concise, coherent narratives of key events (Allan et al., 2001; Tran et al., 2007; Yan et al., 2011; Martschat & Markert, 2017). TLS is crucial for organizing information, enabling efficient access, and supporting downstream tasks such as event tracking and historical analysis. We follow the *Topic TLS* setting (Hu et al., 2024), where given a predefined topic and thousands of related news articles, the system must perform cross-document summarization and event selection to construct a timeline presenting the major milestones.

A central obstacle in TLS is its *evaluation methodology*. Widely used metrics, notably *Date F1* (Chieu & Lee, 2004; Martschat & Markert, 2017) and *A-ROUGE* (Martschat & Markert, 2017; Steen & Markert, 2019), were designed in the pre-LLM era. Their design assumes milestones aligned at the same or nearby timestamps refer to identical events, then measures $n$-gram overlap (e.g., ROUGE) within such alignments. This heuristic was reasonable when automatic semantic

---

[1] Code and data will be released on GitHub upon paper acceptance.
[2] Chapters 1, 2, and 5 have been lightly refined with LLM assistance.

alignment was infeasible, but is inherently shallow: temporal proximity and token overlap are not semantic equivalence. These metrics often diverge from human judgments, especially when abstractive models generate semantically correct but lexically diverse outputs, undervaluing genuine LLM improvements and constraining progress in TLS.

We address this by introducing a new *evaluation framework* leveraging LLM semantic reasoning. Instead of assuming temporal alignment equals semantic equivalence, LLMs directly assess whether two summaries describe the same event. This semantic similarity forms a maximum-weight bipartite matching between predicted and reference events. On this alignment, we design four complementary metrics: (i) **Semantic-Alignment (SA) Score**, quantifying semantic coverage, (ii) **SA Date-F1**, measuring temporal fidelity of semantically matched events, (iii) **SA ROUGE**, assessing textual quality independent of timestamps, and (iv) **STA ROUGE**, requiring both semantic equivalence and temporal consistency. Together, these disentangle semantic matching, temporal accuracy, and linguistic quality, aligning evaluation criteria more closely with human judgement.

To validate the metrics, we extend LLM-TLS into a Full-Stage variant (**FS-LLM-TLS**) that integrates LLMs across the pipeline. We show improvements under existing protocols, and demonstrate that our framework yields more discriminative comparisons against the strong baseline **LLM-TLS** (Hu et al., 2024). Importantly, the proposed metrics better capture the advantages of LLM-driven TLS that traditional evaluation fails to reveal.

Our contributions are:

- A novel **evaluation framework** for TLS grounded in semantic alignment rather than temporal heuristics.
- New **metrics** — SA Score, SA Date-F1, SA ROUGE, STA ROUGE — jointly measuring semantic coverage, temporal fidelity, and textual quality.
- A **FS-LLM-TLS** method used to benchmark these metrics and expose improvements hidden by mainstream protocols.

## 2 RELATED WORK

### 2.1 EVALUATION METRICS FOR TLS

Evaluation is a persistent challenge in TLS. Early work relied on ROUGE (Lin, 2004), which ignores temporal structure (Zhou et al., 2023; Nguyen et al., 2022). Time-aware metrics emerged with Chieu & Lee (2004) and culminated in Martschat & Markert (2017), introducing *Date F1* for timestamp overlap and *A-ROUGE* for overlap after temporal alignment (Steen & Markert, 2019; Ghalandari & Ifrim, 2020). Variants such as Date-Agree ROUGE saw little adoption for being overly strict or inconsistent with human judgment (Tran et al., 2020; Martschat & Markert, 2018; Nguyen & Shirai, 2018). Designed before modern LLMs, these metrics lacked tools for reliable semantic judgment.

Semantic-oriented metrics later emerged. BERTScore (Zhang et al., 2020) measures token-level similarity with contextual embeddings, and QuestEval (Scialom et al., 2021) evaluates whether summaries answer reference-based questions. While advancing semantic evaluation, they were not tailored for TLS and fail to capture both temporal fidelity and semantic coverage. Recent LLM-based methods like G-Eval (Liu et al., 2023) and pairwise event judgments (Walden et al., 2024; Qorib et al., 2024) show stronger alignment with human evaluation.

### 2.2 METHODS FOR TIMELINE SUMMARIZATION

TLS methods have evolved considerably. Early pipelines were extractive, selecting salient sentences or events and ordering them chronologically (Allan et al., 2001; Tran et al., 2007; Yan et al., 2011). Neural and clustering-based methods later improved salience modeling and redundancy reduction (Martschat & Markert, 2017; Ghalandari & Ifrim, 2020), though many baselines still relied on shallow heuristics (Ghalandari & Ifrim, 2020). The rise of LLMs introduced a new direction: Hu et al. (Hu et al., 2024) proposed **LLM-TLS**, using LLMs as semantic oracles to cluster moments into milestones and generate coherent timelines. This has become a strong baseline across datasets, outperforming earlier neural and heuristic systems (Min et al., 2024; Ahmed et al., 2024; Lu et al., 2023; Wu et al., 2024).

## 3 PROBLEM ANALYSE

### 3.1 TEMPORAL HEURISTIC METRICS

Current TLS evaluation framework is largely dominated by two time-aligned metrics: **Date F1** and **A-ROUGE**. The evaluation proceeds in two steps. First, predicted and reference events are aligned strictly by date, and **Date F1** is computed to capture temporal fidelity. Next, **A-ROUGE** computes content overlap conditioned on this temporal alignment by applying ROUGE to summaries of aligned events within a small tolerance window. The exact definitions are given below.

**Date F1.** This metric measures overlap between the sets of predicted and reference dates, ignoring event content. Let $D_{\mathcal{P}}$ and $D_{\mathcal{R}}$ denote the sets of dates in the predicted and reference timelines, respectively. Precision ($P_{\text{date}}$) and Recall ($R_{\text{date}}$) are defined as:

$$P_{\text{date}} = \frac{|D_{\mathcal{P}} \cap D_{\mathcal{R}}|}{|D_{\mathcal{P}}|}, \quad R_{\text{date}} = \frac{|D_{\mathcal{P}} \cap D_{\mathcal{R}}|}{|D_{\mathcal{R}}|}.$$

The Date F1 score is then computed as the harmonic mean:

$$\text{Date-F1} = \frac{2 P_{\text{date}} R_{\text{date}}}{P_{\text{date}} + R_{\text{date}}}.$$

**A-ROUGE.** This metric relaxes exact matching by aligning predicted and reference events according to their nearest dates within a tolerance window $\tau$ (default: $\pm 2$ days in the Tilse toolkit (Hu et al., 2024)). Let $d_i^P$ and $d_j^R$ denote the dates of the $i$-th predicted and $j$-th reference events, respectively, and $y_i^P$, $y_j^R$ their associated summaries. The alignment set is

$$\mathcal{M}_\tau = \{(i,j) \mid |d_i^P - d_j^R| \leq \tau, \; j = \arg\min_k |d_i^P - d_k^R|\}.$$

A-ROUGE$_n$ is then computed as the average ROUGE$_n$ score (Lin, 2004) over aligned pairs:

$$\text{A-ROUGE}_n = \frac{1}{|\mathcal{M}_\tau|} \sum_{(i,j) \in \mathcal{M}_\tau} \text{ROUGE}_n(y_i^P, y_j^R).$$

Both metrics are inherently *time-aligned*. Date F1 ignores semantics entirely, while A-ROUGE assumes temporal proximity implies semantic equivalence. Even with the $\pm 2$ day tolerance, they cannot reliably determine whether two summaries describe the same underlying event, and thus often diverge from human judgments (Martschat & Markert, 2017).

### 3.2 LLM-TLS

Hu et al. (Hu et al., 2024) introduce **LLM-TLS**, a four-stage pipeline that moves from documents to moments, clusters them into events, selects representative sentences, and orders them into milestones to form the timeline. Their central contribution lies in establishing this moment-to-milestone pipeline as the first comprehensive *LLM-based framework* for TLS, and more broadly in framing timeline summarization as a new research paradigm where large language models bridge fine-grained evidence and higher-level narrative milestones.

Despite its contributions, LLM-TLS exhibits several notable limitations. In **Stage 1**, the prompt design emphasizes generating concise news snippets, which often capture only the immediate event while overlooking the real milestone in the background that may be critical for the topic. For instance, in the case of Al Gore from dataset **Entities** (Ghalandari & Ifrim, 2020), a 1999 campaign article may omit his role as Bill Clinton's running mate in 1992, even though this earlier event is an essential part of the timeline. In **Stage 3**, events abstraction reduces each cluster

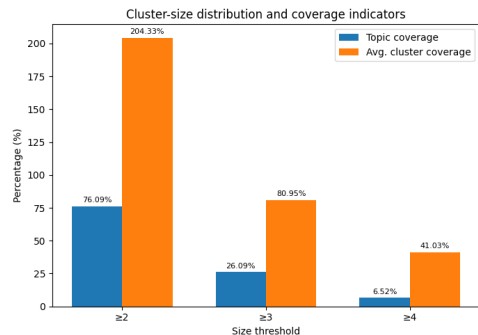

Figure 1: Cluster-size and coverage indicators

to a single representative sentence using traditional ranking algorithms such as TextRank (Mihalcea & Tarau, 2004) or PageRank (Brin & Page, 1998), a process that discards complementary details and produces incomplete summaries. Finally, in **Stage 4**, milestone selection relies purely on statistical frequency, assuming that clusters with more snippets denote more important events. This assumption is problematic: different event types are covered with varying intensity, most clusters contain very few snippets, and human annotators rarely agree that two reports necessarily imply higher salience than one. In practice, many topics do not even contain enough clusters with two or more snippets to fill the required number of milestones, further exposing the weaknesses of frequency-based selection. This phenomenon is shown in Figure 1.

## 4 METHOD

We propose two components: a semantic-alignment evaluation framework and an enhanced LLM-driven pipeline for TLS. The framework introduces **Semantic-Alignment Metrics**, which use semantic consistency as the foundation and evaluate temporal fidelity and textual quality. This ensures milestones are judged more faithfully. To validate the metrics, we extend **LLM-TLS** into **FS-LLM-TLS**, integrating LLMs throughout timeline construction.

### 4.1 SEMANTIC-ALIGNMENT METRICS

#### 4.1.1 DESIGN RATIONALE

Our design refers to the principle of human judgment: people are less tolerant of semantic errors than temporal ones. In long timelines, a shift of a month is acceptable, but a small semantic mismatch means different events. Thus, a milestone must first be semantically aligned. If not, writing quality or timestamp accuracy is irrelevant. Based on this, we calculate SA Score, with SA Date-F1 testing timestamp accuracy and SA-ROUGE testing summary quality. STA-ROUGE combines both, giving the strictest evaluation.

#### 4.1.2 METRIC DEFINITIONS

**Semantic-Alignment Score (SA Score).** To establish a principled foundation for semantic alignment, we design a candidate-based alignment strategy grounded in semantic similarity. Given a predicted timeline $\mathcal{P} = \{(d_i^P, y_i^P)\}_{i=1}^M$ and a reference timeline $\mathcal{R} = \{(d_j^R, y_j^R)\}_{j=1}^K$. Here $d_i^P$ and $d_j^R$ denote the dates of the $i$-th predicted and $j$-th reference events, and $y_i^P$, $y_j^R$ their associated summaries. We first compute pairwise similarity scores.

$$s_{ij} = \text{sim}(y_i^P, y_j^R) \in [0, 1],$$

where $\text{sim}(\cdot)$ is derived from an encoder $\phi(\cdot)$. These scores serve only as soft weights for candidate retrieval. For each reference milestone $y_j^R$, we select the top-$K_{\text{cand}}$ candidates.

$$C_j = \text{TopK}_{i \in [M]} s_{ij}, \quad K_{\text{cand}} = 5,$$

ensuring that only the most semantically relevant predicted events are considered. $K_{\text{cand}}$ is the candidate pool size (default 5), and $\phi(\cdot)$ specifies the similarity backbone.

For each candidate $i \in C_j$, we query the LLM with a yes/no prompt: "Do $y_i^P$ and $y_j^R$ describe the same underlying event?" This yields a binary judgment.

$$\delta_{ij} = \mathbf{1}[\text{LLM-judge}(y_i^P, y_j^R) = \text{Yes}] \in \{0, 1\}.$$

$$\gamma_j = \max_{i \in C_j} \delta_{ij},$$

which indicates whether $y_j^R$ is successfully matched by at least one predicted candidate. The matched set is then

$$\widehat{\mathcal{M}} = \{(i^*, j) \mid \gamma_j = 1, \ i^* = \arg\max_{i \in C_j : \delta_{ij} = 1} s_{ij}\}.$$

Here $\mathcal{M}$ denotes a generic set of candidate index pairs $(i, j)$ between predicted and reference timelines, while $\widehat{\mathcal{M}}$ is the optimized set actually retained after candidate filtering and LLM judgment.

In other words, $\mathcal{M}$ represents the search space of all possible alignments, and $\widehat{\mathcal{M}}$ is the final alignment outcome. Finally, the *SA Score* is computed as the fraction of reference milestones that are semantically covered:

$$\text{SA Score} = \frac{1}{K} \sum_{j=1}^{K} \gamma_j.$$

Here $K$ is the number of reference milestones.

**Semantic-Alignment Date-F1 (SA Date-F1).** Among semantically aligned pairs, we further check whether timestamps coincide. Let $D_{\mathcal{P}}^{\text{match}} = \{d_i^P \mid (i,j) \in \widehat{\mathcal{M}}, d_i^P = d_j^R\}$ and $D_{\mathcal{R}}^{\text{match}} = \{d_j^R \mid (i,j) \in \widehat{\mathcal{M}}, d_i^P = d_j^R\}$. We compute

$$P_{\text{SA-date}} = \frac{|D_{\mathcal{P}}^{\text{match}}|}{|\mathcal{P}|}, \quad R_{\text{SA-date}} = \frac{|D_{\mathcal{R}}^{\text{match}}|}{|\mathcal{R}|},$$

and define

$$\text{SA Date-F1} = \frac{2 P_{\text{SA-date}} R_{\text{SA-date}}}{P_{\text{SA-date}} + R_{\text{SA-date}}}.$$

Note that in our setting $|\mathcal{P}| = |\mathcal{R}|$, so the denominators coincide and precision and recall are always identical; hence, in subsequent tables we report only the unified F1 score for SA Date-F1.

**Semantic-Alignment ROUGE (SA-ROUGE).** Finally, we compute ROUGE-$n$ between summaries of each $(i,j) \in \widehat{\mathcal{M}}$ and average:

$$\text{SA-ROUGE}_n = \frac{1}{|\widehat{\mathcal{M}}|} \sum_{(i,j) \in \widehat{\mathcal{M}}} \text{ROUGE}_n(y_i^P, y_j^R).$$

**Semantic & Temporal-Alignment ROUGE (STA-ROUGE).** To jointly enforce semantic consistency and temporal fidelity, we further introduce STA-ROUGE. Specifically, we restrict ROUGE (Lin, 2004) computation to pairs $(i,j) \in \widehat{\mathcal{M}}$ that are not only judged semantically equivalent by the LLM ($\delta_{ij} = 1$) but also have timestamps within a small tolerance window $\epsilon$, i.e., $|d_i^P - d_j^R| \leq \epsilon$, consistent with A-ROUGE practice. Formally,

$$\text{STA-ROUGE}_n = \frac{1}{|\widehat{\mathcal{M}}_{\text{time}}|} \sum_{(i,j) \in \widehat{\mathcal{M}}_{\text{time}}} \text{ROUGE}_n(y_i^P, y_j^R),$$

where

$$\widehat{\mathcal{M}}_{\text{time}} = \{(i,j) \in \widehat{\mathcal{M}} \mid |d_i^P - d_j^R| \leq \epsilon\}.$$

This metric captures both semantic alignment and correct temporal grounding within tolerance, providing a stricter assessment of timeline summarization quality.

### 4.2 FS-LLM-TLS

Our framework extends the four-stage pipeline of LLM-TLS (Hu et al., 2024) and redefines the role of LLMs for greater efficiency, informativeness, and alignment with human judgment. As shown in Figure 2, the process moves from snippet extraction, where an LLM generates a timestamped event snippet for each document, to event clustering based on semantic similarity and temporal proximity, to cluster-level abstractive summarization that produces a single milestone description, and finally to timeline construction by selecting and ordering $L$ milestones. While retaining the overall structure of LLM-TLS, we redesign three stages—unified snippet extraction, cluster-level abstractive summarization, and hybrid milestone selection—so the LLM operates consistently across the pipeline and aligns with our semantic evaluation metrics.

We redesign how LLMs are deployed in three stages of the pipeline to maximize the potential of LLM in these tasks.

**(i)Unified argument-aware snippet extraction.** For each document the LLM simultaneously extracts structured arguments (*who, what, when*) and generates a grounded 1–2 sentence snippet.

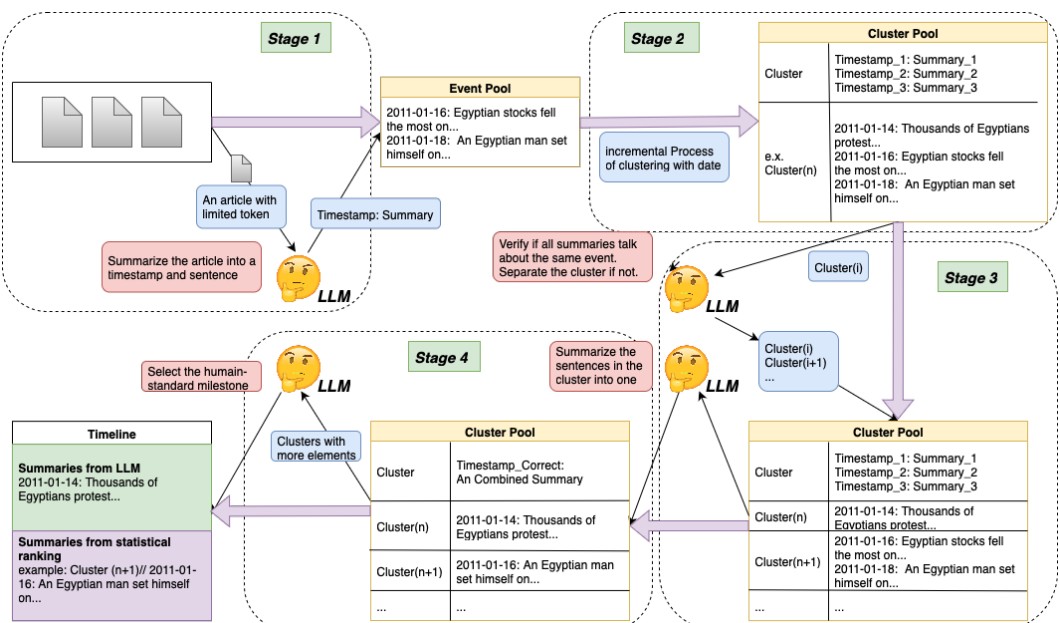

Figure 2: Overview of FS-LLM-TLS pipeline.

This single-prompt formulation preserves background milestones that may be omitted by purely extractive prompts and improves factual consistency between arguments and snippets.

**(ii)Cluster-level abstractive summarization.** Rather than selecting a single representative sentence with TextRank/PageRank, we use the LLM to abstract over *all* snippets in a cluster. By aggregating complementary evidence, the milestone descriptions become more informative and semantically richer, improving recall and coherence.

**(iii)Hybrid milestone selection with batched LLM reasoning.** We first select the top $nL$ clusters by size ($n = 0.5$), then apply a two-pass, batched LLM selection over the remaining candidates (capped at $\approx 4L$) to choose the most important $(1 - n)L$ milestones while staying within context limits.

## 5 EXPERIMENTS

### 5.1 EXPERIMENTAL SETUP

We evaluate our framework on three benchmark datasets: **Entities**, **Crisis**, and **T17**. Following the baseline work of Hu et al. (Hu et al., 2024), we make explicit the data statistics and configurations. The T17 dataset (Tran et al., 2013) consists of 9 predefined topics and spans roughly 7 months of real-world news data. The Crisis dataset (Rajaby Faghihi et al., 2022) encompasses 4 topics, similarly covering long-span armed conflict or crisis news. The Entities dataset (Ghalandari & Ifrim, 2020) is larger, containing 47 distinct topics, with time-ranges extending up to 12 years.

Following previous work, we report three primary metrics: (1) **Date F1** for temporal alignment, (2) **AR1** (A-ROUGE-1), and (3) **AR2** (A-ROUGE-2). Martschat & Markert (2017) further proposed stricter variants such as *date-agreement ROUGE* as well as other alignment-based measures, but these have rarely been adopted in subsequent TLS work due to their overly severe penalization or limited interpretability. In addition, to reflect human perspectives more directly, we employ two supplementary metrics: (4) **CE** (Content Equivalence), which counts the number of semantically consistent milestones between predicted and reference timelines regardless of timestamps, and (5) **DACE** (Date-Agreement Content Equivalence), which further requires both semantic consistency and temporal correctness. These metrics approximate how humans judge whether key events are correctly captured and temporally faithful.

Unless otherwise specified, we use the `LLaMA2-13B-Instruct` model throughout all experiments. For reproducibility, the prompt templates for each stage of our pipeline, the model hyperparameters, and detailed configuration of all LLMs are provided in the Appendix.

## 5.2 OVERALL RESULTS UNDER MAINSTREAM AND OUR METRICS

This experiment evaluates the effectiveness of our FS-LLM-TLS method compared with the baseline across three benchmark datasets: **Entities**, **Crisis**, and **T17**. As shown in Table 1, we report results under two complementary metrics frameworks and human perspectives.

Table 1: Overall performance comparison of metrics and method

| Dataset | Method | Standard Metrics | | | Semantic-Alignment Metrics | | | | | Human Perspective | |
|---------|--------|---------|-----|-----|----------|-----------|-------|-------|--------|-------|-------|
| | | Date F1 | AR1 | AR2 | SA-Score | SA-Date F1 | SA-R1 | SA-R2 | STA-R1 | CE | DACE |
| Entities | Baseline | 0.232 | 0.091 | 0.037 | 0.281 | 0.152 | 0.115 | 0.049 | 0.086 | 6.31 | 3.36 |
| Entities | Ours (n=0.5) | **0.256** | **0.095** | **0.040** | **0.330** | **0.172** | **0.130** | **0.061** | **0.099** | **7.98** | **4.38** |
| Crisis | Baseline | 0.530 | 0.312 | 0.156 | 0.644 | 0.488 | 0.338 | 0.170 | 0.277 | 89.74 | 66.27 |
| Crisis | Ours (n=0.5) | **0.545** | **0.325** | **0.163** | **0.680** | **0.535** | **0.358** | **0.182** | **0.299** | **97.92** | **77.04** |
| T17 | Baseline | 0.380 | 0.267 | 0.121 | 0.482 | 0.404 | 0.352 | 0.062 | 0.237 | 36.74 | 32.17 |
| T17 | Ours (n=0.5) | **0.410** | **0.280** | **0.130** | **0.523** | **0.458** | **0.392** | **0.074** | **0.269** | **48.12** | **41.03** |

Across all datasets, FS-LLM-TLS outperforms the baseline under both Standard and Semantic-Alignment Metrics. Standard Metrics improve modestly (+4–6%), reflecting limited sensitivity to semantic reasoning. SA metrics show larger gains (+10–16%), with SA-Score strongest, and their consistency aligns more closely with human perspectives. This indicates that date-aligned metrics compress differences in event selection and sentence quality, while SA metrics reveal the true performance gap. The advantage stems from FS-LLM-TLS strengthening semantic completeness and milestone selection, where LLMs excel. Thus, SA metrics capture the core challenge of TLS—event selection quality—and amplify differences between FS-LLM-TLS and the baseline. On the heterogeneous **Entities** dataset, SA-Score improves by +24.6% versus only +10.3% with Date F1, confirming its superior discriminative power.

## 5.3 INFLUENCE ANALYSIS OF STAGE-WISE OPTIMIZATION ON METRICS AND ABLATION STUDY

To assess the contribution to the metrics of each optimization step in FS-LLM-TLS, we conduct a comprehensive ablation study on the **Entities** dataset. This experiment isolates the effect of individual stage-level improvements as well as their combinations, thereby illustrating how different components interact within the full pipeline. The findings is shown in Table 2.

Table 2: Influence analysis of stage-wise optimization on metrics and ablation study

| Method | Standard Metrics | | | Semantic-Alignment Metrics | | | | |
|--------|---------|-----|-----|----------|-----------|-------|-------|--------|
| | Date F1 | AR1 | AR2 | SA-Score | SA-Date F1 | SA-R1 | SA-R2 | STA-R1 |
| Baseline | 0.232 | 0.091 | 0.037 | 0.281 | 0.152 | 0.115 | 0.049 | 0.086 |
| FS-LLM-TLS | **0.256** | **0.095** | **0.040** | **0.330** | **0.172** | **0.130** | **0.061** | **0.99** |
| Baseline + S1 | 0.230 | 0.092 | 0.037 | 0.285 | 0.154 | 0.116 | 0.050 | 0.086 |
| Baseline + S3 | 0.232 | 0.094 | 0.040 | 0.285 | 0.152 | 0.121 | 0.052 | 0.089 |
| Baseline + S4 | 0.253 | 0.093 | 0.039 | 0.332 | 0.175 | 0.127 | 0.057 | 0.095 |
| Baseline + S1+S3 | 0.236 | 0.095 | 0.040 | 0.288 | 0.156 | 0.123 | 0.055 | 0.95 |
| Baseline + S3+S4 | 0.252 | 0.094 | 0.039 | 0.346 | 0.177 | 0.132 | 0.058 | 0.98 |
| Baseline + S1+S4 | 0.249 | 0.093 | 0.039 | 0.332 | 0.175 | 0.131 | 0.058 | 0.097 |

Overall, the ablation shows that both individual optimizations and their combinations improve performance. Only methods with Stage 4 yield clear gains under Semantic-Alignment metrics, especially SA-Score (+21.7%), while Date F1 rises more modestly (+9.1%). Stage 4's LLM-based

selection identifies true "milestones," whereas Stage 1 and Stage 3 mainly affect sentence quality, leaving semantic alignment largely unchanged. These results highlight the gap between token-level ROUGE and semantic similarity. SA metrics provide a necessary complement to traditional time-prior metrics: SA-Score captures milestone selection, SA-Date F1 temporal accuracy, and SA-ROUGE sentence quality. Improvements of +26.3%, +24.5%, and +24.6% in SA-Date F1, SA-R2, and SA-Score confirm FS-LLM-TLS's advantages in temporal accuracy, sentence quality, and event selection.

## 5.4 Analysis on Discriminative Power of SA Metrics

In the previous experiment, we demonstrated that SA metrics are primarily influenced by the quality of event selection (the task of Stage 4). Building on this finding, in the following experiment we focus exclusively on Stage 4 across all three datasets. In each case, we vary the proportion of milestones in the predicted timeline that are selected through LLM-based filtering: 100% (n=0), 75% (n=0.25), and 50% (n=0.5).

Table 3: Influence on metrics of LLM's participation on Stage 4 across datasets

| Dataset | Method | Mainstream Metrics | | | SA Metrics | | | | | Human Perspective | |
|---|---|---|---|---|---|---|---|---|---|---|---|
| | | Date F1 | A-R1 | A-R2 | SA-Score | SA-Date F1 | SA-R1 | SA-R2 | STA-R1 | CE | DACE |
| Entities | Baseline | 0.232 | 0.091 | 0.037 | 0.281 | 0.152 | 0.115 | 0.049 | 0.086 | 6.31 | 3.36 |
| Entities | S4 (n=0) | 0.192 | 0.088 | 0.035 | 0.296 | 0.148 | 0.121 | 0.052 | 0.088 | 6.73 | 3.37 |
| Entities | S4 (n=0.25) | 0.243 | 0.094 | 0.039 | 0.338 | 0.192 | 0.132 | 0.060 | 0.099 | 7.70 | 4.38 |
| Entities | S4 (n=0.5) | 0.256 | 0.095 | 0.040 | 0.350 | 0.192 | 0.135 | 0.061 | 0.102 | 7.98 | 4.38 |
| Crisis | Baseline | 0.530 | 0.312 | 0.156 | 0.644 | 0.488 | 0.338 | 0.170 | 0.277 | 89.74 | 66.27 |
| Crisis | S4 (n=0) | 0.485 | 0.284 | 0.143 | 0.636 | 0.408 | 0.314 | 0.158 | 0.259 | 91.58 | 58.76 |
| Crisis | S4 (n=0.25) | 0.538 | 0.319 | 0.160 | 0.674 | 0.528 | 0.352 | 0.178 | 0.294 | 97.30 | 76.43 |
| Crisis | S4 (n=0.5) | 0.545 | 0.325 | 0.163 | 0.680 | 0.535 | 0.358 | 0.182 | 0.299 | 97.92 | 77.04 |
| T17 | Baseline | 0.380 | 0.267 | 0.121 | 0.482 | 0.385 | 0.352 | 0.062 | 0.237 | 36.74 | 32.17 |
| T17 | S4 (n=0) | 0.388 | 0.270 | 0.124 | 0.508 | 0.412 | 0.379 | 0.069 | 0.260 | 46.74 | 37.90 |
| T17 | S4 (n=0.25) | 0.423 | 0.288 | 0.135 | 0.530 | 0.448 | 0.396 | 0.075 | 0.270 | 48.76 | 41.22 |
| T17 | S4 (n=0.5) | 0.410 | 0.280 | 0.130 | 0.523 | 0.446 | 0.392 | 0.074 | 0.269 | 48.12 | 41.03 |

For milestone selection ratios of 75% and 50%, improvements appear across all metrics, with relative strengths varying by dataset. When milestones are 100% LLM-selected (n=0), Date F1 drops, yet SA-Score remains stable in **Entities** (+5.3%) and even rises in **T17** (+5.4%), while STA-R1 also increases (+2.3%). This shows that LLMs still identify meaningful events, though by different principles. In **Entities**, CE improves by +6.7%, +22.0%, and +26.5% for n=0, 0.25, and 0.5, while DACE changes by −3.3%, +30.4%, and +30.4%. Correspondingly, SA-Date F1 rises by −2.6%, +26.3%, and +26.3%, and SA-Score by +5.3%, +20.3%, and +24.6%. These patterns show SA-Date F1 aligns with DACE, while SA-Score tracks CE, confirming that semantic metrics reflect human judgments more reliably than date-only metrics. Across datasets, SA-Score under n=0 decreases slightly in **Crisis** (−1.2%), increases in **T17** (+5.4%), and stays stable in **Entities** (+5.3%), suggesting that statistical heuristics remain useful but should not dominate evaluation. Relying only on Date F1 and A-ROUGE would suggest that LLM-only selection underperforms. SA metrics reveal the opposite: declines come from temporal misalignment, while semantic matching and linguistic quality improve.

## 5.5 Robustness

To test robustness, we evaluate on medium-scale models of similar size: Qwen3-14B and LLaMA-2-13B. This cross-model setting checks whether improvements and SA metrics remain consistent across architectures.

The results show that our method consistently improves over the baseline under both models. Moreover, the Semantic-Alignment metrics exhibit stable and coherent behavior across different architectures, reinforcing the robustness and generalizability of our evaluation framework.

Table 4: Model-agnostic evaluation on medium-scale LLMs (13B–14B).

| Method | Model | Standard Metrics | | | Semantic-Alignment Metrics | | | | |
|---|---|---|---|---|---|---|---|---|---|
| | | Date F1 | AR1 | AR2 | SA-Score | SA-Date F1 | SA-R1 | SA-R2 | STA-R1 |
| Baseline | LLaMA-2-13B | 0.232 | 0.091 | 0.037 | 0.281 | 0.152 | 0.115 | 0.049 | 0.086 |
| Ours | LLaMA-2-13B | **0.256** | **0.095** | **0.040** | **0.330** | **0.172** | **0.130** | **0.061** | **0.99** |
| Baseline | Qwen-14B | 0.230 | 0.090 | 0.037 | 0.285 | 0.152 | 0.114 | 0.048 | 0.085 |
| Ours | Qwen-14B | 0.252 | 0.095 | 0.040 | 0.334 | 0.180 | 0.131 | 0.060 | 0.094 |

## 5.6 SCALABILITY WITH LARGER MODELS

To test scalability, we compare FS-LLM-TLS with the baseline across LLaMA-2 and Qwen3 models of different sizes, using the **Entities** dataset for consistency.

Table 5: Scalability analysis with larger LLaMA2 models on **Entities**.

| Method | Model | Standard Metrics | | | SA Metrics | | | | |
|---|---|---|---|---|---|---|---|---|---|
| | | Date F1 | AR1 | AR2 | SA-Score | SA-Date F1 | SA-R1 | SA-R2 | STA-R1 |
| Baseline | LLaMA-2-13B | 0.232 | 0.091 | 0.037 | 0.281 | 0.152 | 0.115 | 0.049 | 0.086 |
| Ours | LLaMA-2-13B | 0.256 | 0.095 | 0.040 | 0.330 | 0.172 | 0.130 | 0.061 | 0.099 |
| Baseline | LLaMA-2-70B | 0.236 | 0.093 | 0.039 | 0.281 | 0.152 | 0.119 | 0.050 | 0.088 |
| Ours | LLaMA-2-70B | 0.262 | 0.099 | 0.044 | 0.352 | 0.175 | 0.140 | 0.068 | 0.105 |
| Baseline | Qwen-14B | 0.230 | 0.090 | 0.037 | 0.285 | 0.152 | 0.114 | 0.048 | 0.085 |
| Ours | Qwen-14B | 0.252 | 0.095 | 0.040 | 0.334 | 0.180 | 0.131 | 0.060 | 0.094 |
| Baseline | Qwen3-30B | 0.234 | 0.093 | 0.039 | 0.280 | 0.154 | 0.116 | 0.050 | 0.087 |
| Ours | Qwen3-30B | 0.258 | 0.096 | 0.042 | 0.348 | 0.176 | 0.138 | 0.066 | 0.104 |
| Baseline | Qwen3-235B | 0.236 | 0.095 | 0.040 | 0.289 | 0.159 | 0.121 | 0.050 | 0.088 |
| Ours | Qwen3-235B | **0.273** | **0.101** | **0.047** | **0.376** | **0.195** | **0.148** | **0.070** | **0.109** |

Results show that FS-LLM-TLS scales well with model capacity. Date F1 gains are modest (+11.1%), while SA-Score improves more substantially (+21.5%) and SA-Date F1 moderately (+13.4%). This indicates that larger models mainly strengthen event selection rather than temporal alignment. In contrast, the baseline benefits little from scaling, with only +1.7% in Date F1 and +2.8% in SA-Score. Thus, deeper integration of larger LLMs into the TLS pipeline makes SA metrics essential for exposing real improvements and limitations.

## 6 CONCLUSION

We re-examine TLS evaluation in the era of strong models. Existing metrics like Date F1 and A-ROUGE conflate temporal proximity with semantics, masking real progress. We introduce a semantic-alignment framework with **SA metrics** that separates semantic coverage, temporal fidelity, and textual quality. To validate them, we propose **FS-LLM-TLS**, a stage-wise pipeline integrating LLM reasoning in snippet extraction, cluster abstraction, and milestone selection. Gains under traditional metrics are modest, but SA metrics reveal clear advances in event selection and completeness. SA metrics thus offer stronger discriminative power and closer alignment with human judgment, providing a principled framework for future TLS evaluation.

## 7 LIMITATIONS AND FUTURE WORK

SA metrics still operate at the sentence level and use discrete temporal tags. Future work could extend them to discourse-aware or cross-document alignment and involve human-in-the-loop evaluation. FS-LLM-TLS improves event selection and completeness, but temporal accuracy gains are limited. Models detect salient events well but struggle with precise dating. Addressing this may require combining statistical priors with model reasoning for temporal grounding.

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

## A    PSEUDOCODE

Other content will be released upon paper acceptance.

## B    PROMPT TEMPLATES

Content will be released upon paper acceptance.

## C    MODEL AND ENVIRONMENT SETTINGS

Content will be released upon paper acceptance.

## D    ADDITIONAL EXPERIMENTS

Content will be released upon paper acceptance.

**Algorithm 1** One-shot Argument-aware Snippet Extraction

---

**Require:** Documents $\mathcal{D} = \{d_1, \ldots, d_N\}$; LLM $\mathcal{M}$; prompt template $\mathsf{P}$
**Ensure:** For each $d_i$: structured arguments $(a_i^{who}, a_i^{what}, a_i^{when})$ and snippet $s_i$
1: **for** $i = 1$ to $N$ **do**
2: $\quad$ `prompt` $\leftarrow \mathsf{P}(d_i)$
3: $\quad (a_i^{who}, a_i^{what}, a_i^{when}, s_i) \quad\quad \leftarrow$ $\mathcal{M}(\texttt{prompt})$
4: **end for**
5: **return** $\{(a_i^{who}, a_i^{what}, a_i^{when}, s_i)\}_{i=1}^N$

---

**Algorithm 2** Cluster-level Abstractive Summarization

---

**Require:** Clusters $\mathcal{C} = \{C_1, \ldots, C_K\}$; LLM $\mathcal{M}$; prompt template $\mathsf{P}$; decoding temperature $\tau$
**Ensure:** For each $C_i$: abstractive milestone $S_i$ and representative timestamp $t_i$
1: **for** $i = 1$ to $K$ **do**
2: $\quad U_i \leftarrow \text{SelectAll}(C_i)$ $\quad \triangleright$ use all snippets in $C_i$
3: $\quad$ `prompt` $\leftarrow \mathsf{P}(U_i)$
4: $\quad S_i \leftarrow \mathcal{M}(\texttt{prompt}; \tau)$
5: $\quad t_i \leftarrow \text{ModeDate}(C_i)$
6: **end for**
7: **return** $\{(t_i, S_i)\}_{i=1}^K$

---

Figure 3: (Left) Snippet extraction with unified argument and summary generation. (Right) Cluster-level abstractive summarization.

