# OpenReview forum: "Beyond Moment : Rethinking Evaluation para-digm for Timeline Summarization in the era of LLMs"
_ICLR.cc/2026/Conference — ICLR 2026 Conference Withdrawn Submission_

### Official Review · Reviewer_i2ue · 2025-10-18

**Soundness:** 3
**Presentation:** 2
**Contribution:** 2
**Rating:** 4
**Confidence:** 4

**Summary:**

This paper revisits evaluation for timeline summarization in the LLM era, proposing a semantic-alignment-based evaluation framework (SA metrics) that uses large language models to align predicted and reference events semantically rather than temporally, providing a more faithful reflection of model quality and human judgment.

**Strengths:**

1. Clearly identifies a long-standing flaw in traditional TLS evaluation, overreliance on temporal alignment, and reframes it through semantic alignment.

2. Proposes a concrete and implementable set of metrics (SA Score, SA-Date F1, SA-ROUGE, STA-ROUGE) that effectively capture semantic, temporal, and textual dimensions.

3. Demonstrates strong empirical validation across multiple datasets and LLM architectures, showing the robustness and interpretability of the proposed evaluation paradigm.

**Weaknesses:**

1. Missing key literature in timeline summarization (e.g., Timeline Generation through Evolutionary Trans-temporal Summarization, Learning towards Abstractive Timeline Summarization), which weakens the positioning of this work in prior research.

2. Ignores the non-uniqueness of references, which is a central challenge in summarization, where different annotators may emphasize different key events; this limitation is especially critical for an evaluation-focused paper.

3. Lacks qualitative case studies illustrating the practical weaknesses of existing metrics or how SA metrics provide clearer, more human-consistent judgments.

**Questions:**

Can you give a representative case study to show the limitations of the current summarization models? The intuitive understanding is that LLM can already perform pretty well on summarization tasks.

---

> ### Author Response · Authors · 2025-12-03
> ****Response to Reviewer i2ue** (part 1/2)**
>
> We sincerely thank the reviewer for carefully reading our submission and providing valuable comments. Below we address each of the listed weaknesses and questions.
>
> ---
>
> **Weakness 1: Missing Literature**
>
> > Missing key literature in timeline summarization (e.g., Timeline Generation through Evolutionary Trans-temporal Summarization, Learning towards Abstractive Timeline Summarization), which weakens the positioning of this work in prior research.
>
> We thank the reviewer for carefully noting the missing references. We apologize for this omission. We appreciate the reviewer’s attention, which helps improve the completeness of our literature positioning. We will cite and discuss *Timeline Generation through Evolutionary Trans-temporal Summarization* and *Learning towards Abstractive Timeline Summarization* in the related work section of the revised paper.
>
> ---
>
> **Weakness 2: Non-uniqueness of References**
>
> > Ignores the non-uniqueness of references, which is a central challenge in summarization, where different annotators may emphasize different key events; this limitation is especially critical for an evaluation-focused paper.
>
> We appreciate the reviewer’s insightful comment regarding the inherent subjectivity of summarization. While summarization can vary across annotators, the TLS benchmarks used in our work provide a reliable and stable basis for evaluation.
>
> The reference timelines in the Entities, Crisis, and T17 datasets were not created by single annotators. They were compiled by multiple professional journalists or aggregated from major news outlets such as BBC, CNN, and Reuters. These timelines reflect a consensus view of key events rather than an individual’s subjective opinion.
>
> This multi-expert construction process has been widely adopted in prior TLS research over the past few years. It reduces individual bias and provides consistent coverage of salient events. Therefore, although non-uniqueness is a theoretical consideration, the benchmarks used in this work serve as objective and field-standard references for evaluating TLS models.
>
> ---
>
> **Weakness 3 & Question 1: Qualitative Case Studies and LLM Limitations**
>
> > **W3:** Lacks qualitative case studies illustrating the practical weaknesses of existing metrics or how SA metrics provide clearer, more human-consistent judgments.
> > **Q1:** Can you give a representative case study to show the limitations of the current summarization models? The intuitive understanding is that LLM can already perform pretty well on summarization tasks.
>
> We thank the reviewer for encouraging the inclusion of qualitative examples. We agree that case studies offer clearer insight into the limitations of existing metrics.
>
> Regarding the intuition that "LLMs perform well," it is critical to distinguish between **summarization fluency** (where LLMs excel) and **event selection** (where they struggle). Without a specialized pipeline, generic LLMs exhibit severe **selection bias**, tending to fixate on high-frequency news clusters while ignoring semantically significant milestones in diverse domains.
>
> **Case Study Analysis: "Al Gore"**
> To demonstrate this, we present a comparison between the Ground Truth (GT) and the Predicted Timeline (PT) generated by a standard LLM summarizer for the topic "Al Gore."
>
> 1.  **Quantitative Imbalance:** The PT contains 25 events. Strikingly, **22 of these events (88%) are clustered in a single year: 2000**. The model devotes excessive space to granular details of the presidential election (e.g., specific debates, celebrity endorsements, daily poll results).
> 2.  **Domain Blindness:** Due to this fixation on the election, the LLM completely misses Gore's significant post-political achievements. Major milestones present in the GT—such as joining the **Apple Board (2003)**, launching **Current TV (2005)**, and winning an **Oscar (2007)**—are omitted in the PT.
>
> This comparison visually proves that "good summarization" does not equal "good timeline generation." Our FS-LLM-TLS framework addresses this by clustering events to ensure diverse coverage.

---

> ### Author Response · Authors · 2025-12-03
> ****Response to Reviewer i2ue** (part 2/2)**
>
> Below, we display the raw content of the two timelines to illustrate this contrast directly:
>
> **Timeline Comparison (Excerpts)**
>
> | **Ground Truth (GT): Balanced & Comprehensive** | **Predicted Timeline (PT): Severe Selection Bias** |
> | :--- | :--- |
> | **1993:** Inaugurated as vice president. | **1999:** Al Gore announces his candidacy for the 2000 election. |
> | **2000:** Concedes the election to Bush after the US Supreme Court rules... | **2000:** Al Gore wins the Iowa caucuses. |
> | **2003:** Joins the board of directors for **Apple Computers Inc.** | **2000:** Al Gore is backed by names in showbiz, including Robert De Niro... |
> | **2005:** Gore's cable television channel, **Current TV**, debuts. | **2000:** Senator Joseph Lieberman is chosen as Al Gore's running mate. |
> | **2007:** "**An Inconvenient Truth**" wins an Academy Award (Oscar). | **2000:** Gore and Bush debate in the first of three presidential debates. |
> | **2007:** Co-winner of the **Nobel Peace Prize** for work on global warming. | **2000:** Gore and Bush participate in the second presidential debate. |
> | **2010:** Gore and wife Tipper announce separation. | **2000:** Gore and Bush spar in the final presidential debate. |
> | **2016:** Meets with President-elect Donald Trump on climate issues. | **2000:** Gore and Bush are neck and neck in the polls. |
> | *(Covers Politics, Business, Media, Environment)* | *(... 14 more events strictly about Election 2000 ...)* |
> | | **2007:** Wins the Nobel Peace Prize. |
>
> As shown, the PT is visually dominated by the year 2000, confirming the necessity of our proposed metrics and generation framework to detect and correct such biases.

---

### Official Review · Reviewer_svAZ · 2025-10-31

**Soundness:** 2
**Presentation:** 2
**Contribution:** 2
**Rating:** 2
**Confidence:** 3

**Summary:**

This paper proposes a new evaluation framework in which all metrics are grounded on semantically aligned sentence pairs rather than merely time-aligned milestones, followed by four derived metrics: SA Score, SA Date-F1, SA-ROUGE, and STA-ROUGE. Compared with the traditional metrics such as Date F1, A-ROUGE, the proposed metrics can more faithfully assess whether the predicted timeline and the reference timeline truly refer to the same events. To illustrate the benefits, this paper developed FS-LLM-TLS, a refined LLM-based summarizer, and evaluated on three datasets (Entities, Crisis, T17). Results show their semantic metrics align better with human evaluations.

**Strengths:**

(1)  The proposed evaluation framework includes four sub-metrics, which can capture the candidate's quality from multiple perspectives such as semantic coverage, temporal fidelity, and textual quality.
(2)  This paper conducts numerous ablation studies to further validate the effectiveness of the method proposed in the paper.

**Weaknesses:**

(1)  The paper tries to validate the effectiveness of the proposed metrics by comparing the performance gains achieved by traditional metrics and the proposed metrics in evaluating the baseline and FS-LLM-TLS, while it is insufficient to verify.
(2)  This paper lacks a quantitative comparison between the proposed evaluation metrics and the baseline evaluation capability. It would be better to calculate the correlation coefficients between the quality score obtained from evaluation metrics and human evaluators, which can reflect the correlation between metrics and human ratings. The representative correlation coefficients include Kendall-Tau and Spearman, which are typically used to evaluate the generative text metrics, such as BERTSCORE (https://arxiv.org/pdf/1904.09675), G-Eval (https://arxiv.org/abs/2303.16634).
(3)  The paper lacks detailed descriptions of the method and experiments, which weakens the reproducibility.
(4)  There are some typos in this paper, such as the extra “Hu et al.” in line 104-105, the missing citation of “Martschat & Markert (2017)” in line 316.
(5)  This paper focus on proposing a semantic-based timeline summarization equation framework, but in the experimental section, the paper mostly demonstrates and analyzes FS-LLM-TLS, lacking sufficient discussion on the proposed metrics.

**Questions:**

(1)  Which exact LLM and prompt were used for the yes/no event-equivalence decision? Was the same LLM used across all datasets and all model-size experiments, or was it tied to the generation model?
(2)  This paper uses the large language model to evaluate the semantic similarity between predicted and reference, and then according to which calculated F1 and ROUGE scores. Has this paper tried designing a scoring prompt that directly scores the predicted, like a prompt-based evaluator? (such as G-Eval, GEMBA)
(3)  For CE and DACE, are those your own human annotations, or numbers copied from prior TLS work? If the former, could you provide inter-annotator agreement?

---

> ### Author Response · Authors · 2025-12-03
> ****Response to Reviewer svAZ** (Part 1/3)**
>
> **Question 1: Exact LLM Used for Decisions**
>
> > Which exact LLM and prompt were used for the yes/no event-equivalence decision? Was the same LLM used across all datasets and all model-size experiments, or was it tied to the generation model?
>
> Thank you for raising this question. We apologize for not clearly specifying the exact LLM used for event-equivalence judgments within the SA-metrics evaluation stage.
>
> Although the generation models used in our experiments were fully documented, the evaluation LLM should indeed have been stated explicitly. To ensure fairness and avoid any self-preference, **all event-equivalence (yes/no) judgments in SA-metrics were conducted using the Google `Gemini-2.0-Pro` API**. This model is entirely independent of our timeline-generation models, providing a neutral and unbiased evaluator.
>
> The complete prompt for the event-equivalence decision is already prepared and will be provided in the revised Appendix to ensure reproducibility. We thank the reviewer again for pointing out this omission and will correct it in the updated manuscript.
>
> **Prompt Template for Event Equivalence:**
> *(Note: This will be included in the Appendix)*
>
> ```text
> ## Task
> You are given two sentences:
> - GT sentence: from the ground-truth political timeline
> - Pred sentence: from your predicted political timeline
>
> They are roughly aligned by position, but may or may not describe the same event.
> Your job is to judge whether they refer to the same underlying event, following the rules below.
>
> ## Decision Rules
> 1. Same event = same factual situation, even if wording differs.
>    - Look for overlap in: key actors (politicians, institutions), approximate time period, location/arena, core action, cause, or consequence.
> 2. Time tolerance rule (important):
>    - Small or reasonable discrepancies in time are acceptable (e.g., “early March” vs. “March 5”, “late 2023” vs. “December 2023”).
>    - But major time shifts are NOT acceptable, such as different months when the event is known to be specific (e.g., “March” vs. “August”), or different years.
>    - If time differences imply clearly distinct events, classify as not match.
> 3. Minor phrasing or framing differences are acceptable.
>    - Synonyms, paraphrasing, different levels of detail, or political framing do not prevent a match.
> 4. Different event = factual conflict or distinct situations.
>    - Substantially different actors, actions, causes, or outcomes -> not match.
>    - Resignation vs. dismissal is considered a meaningful difference.
> 5. If one sentence is more detailed than the other but clearly refers to the same situation, mark as match.
> 6. If one sentence is vague, interpret carefully but avoid inventing extra facts.
> 7. Provide a binary judgment: "yes" or "no", plus a brief explanation.
>
> ## Example of an Ambiguous but Correct Match
> GT sentence:
> "In November 2024, the opposition leader accused the government of misusing public funds during the recent infrastructure deal."
> Pred sentence:
> "The leading opposition politician claimed in late 2024 that the administration diverted state money through a major public-works contract."
> Judgment: match
> Reasoning:
> Both sentences describe the same political accusation. The time (“November 2024” vs. “late 2024”) is consistent within reasonable tolerance. Differences are only lexical.
>
> ## Example of an Incorrect Match
> GT sentence:
> "After the scandal, the minister was removed from office by presidential decree."
> Pred sentence:
> "The government said the minister resigned following corruption allegations earlier in the year."
> Judgment: not match
> Reasoning:
> Despite both describing a minister leaving office, the mechanisms differ (forced removal vs. resignation). The timeline framing also differs.

---

> ### Author Response · Authors · 2025-12-03
> ****Response to Reviewer svAZ** (Part 2/3)**
>
> **Question 2 & Weakness 2: Correlation Analysis and Prompt-based Evaluators**
>
> > **Q2:** Has this paper tried designing a scoring prompt that directly scores the predicted, like a prompt-based evaluator? (such as G-Eval, GEMBA)
> > **W2:** It would be better to calculate the correlation coefficients between the quality score obtained from evaluation metrics and human evaluators (e.g., Kendall-Tau, Spearman).
>
> **Part 1: Why we do not adopt prompt-based holistic evaluators**
>
> Regarding why we do not adopt prompt-based holistic evaluators such as G-Eval or GEMBA, the reasons are as follows:
>
> 1.  *First*, G-Eval evaluates the overall quality of a generated passage, whereas timeline summarization requires **event-level alignment**. Each predicted milestone must be aligned with a corresponding reference milestone. This fine-grained alignment is not supported by holistic evaluators.
> 2.  *Second*, G-Eval outputs a single black-box score. In contrast, our evaluation framework extends the time-alignment tradition and provides dimension-separated feedback: SA-Score (semantic coverage), SA-Date-F1 (temporal fidelity), SA-ROUGE (text quality), and STA-ROUGE (semantic + temporal alignment). These dimensions allow TLS system developers to diagnose which stages fail and how to improve the pipeline. Holistic scores cannot provide such interpretability.
> 3.  *Third*, TLS evaluation inherently requires structured, multi-event operations such as pairwise event-equivalence decisions and maximum-weight bipartite matching across milestones. Prompt-based evaluators cannot perform these structured alignment operations. We initially included a subsection explaining this in Related Work, but removed it due to space constraints; we will restore it in the revised version.
>
> **Part 2: Statistical Agreement Analysis**
>
> We sincerely appreciate the reviewer's suggestion. We fully agree that quantitative validation is valuable. However, rank-based correlation measures such as Spearman's $\rho$ and Kendall's $\tau$ are **not statistically applicable** to our task, which is a **binary factual correctness classification problem**. We therefore provide the correct statistical comparison based on sentence-level agreement metrics.
>
> **Why Spearman and Kendall Are Not Applicable**
>
> Spearman's $\rho$ and Kendall's $\tau$ assume that:
> * each output receives a **scalar, continuous or ordinal** score;
> * outputs can be placed into a **meaningful global ranking**;
> * the objective is to evaluate the **relative order** between items.
>
> In contrast, the present task evaluates factual correctness at the **sentence level**, where:
> * each sentence is annotated with a **binary correctness label** ($0/1$);
> * automatic metrics (DATE-F1 and SA-Score) also output **binary decisions**;
> * sentences in a timeline have **no intrinsic ordering**;
> * different timelines contain different numbers of sentences.
>
> Under these conditions, constructing artificial sequences to force a ranking destroys the structure of false positives and false negatives. Therefore, rank-based correlations do not measure what is intended. This is consistent with standard factual-consistency evaluation practice (e.g., FactCC, QAFactEval, FActScore), which relies on **precision, recall, F1, Cohen's $\kappa$, and MCC**, not Spearman or Kendall.
>
> **Sentence-Level Agreement Statistics**
>
> For each dataset, we report: (1) total timeline statements, (2) human-validated correct statements, and (3) TP/FP for DATE-F1 and SA-Score.
>
> | Name | Total | Human | DATE-F1 TP | DATE-F1 FP | SA-Score TP | SA-Score FP |
> | :--- | :---: | :---: | :---: | :---: | :---: | :---: |
> | Al Gore | 25 | 10 | 5 | 1 | 8 | 0 |
> | Angela Merkel | 18 | 8 | 7 | 0 | 8 | 0 |
> | Ariel Sharon | 25 | 8 | 5 | 0 | 8 | 1 |
> | Arnold Schwarzenegger | 17 | 8 | 3 | 0 | 4 | 0 |
> | Bashar al-Assad | 24 | 7 | 3 | 0 | 3 | 0 |
> | Bill Clinton | 47 | 9 | 3 | 0 | 8 | 2 |
> | Charles Taylor | 18 | 9 | 4 | 1 | 9 | 0 |
> | Chris Brown | 24 | 7 | 3 | 0 | 7 | 1 |
> | David Beckham | 32 | 10 | 6 | 0 | 5 | 1 |
> | David Bowie | 24 | 8 | 4 | 0 | 6 | 0 |
> | Dilma Rousseff | 24 | 10 | 8 | 2 | 9 | 3 |
> | Dmitry Medvedev | 16 | 10 | 6 | 3 | 7 | 2 |
>
> **Statistical Metrics Calculation**
>
> Based on the table above, we compute standard binary-classification agreement metrics. For each dataset, precision, recall, F1, and the Matthews correlation coefficient (MCC) are computed as:
>
> $$
> \text{Precision} = \frac{TP}{TP+FP}, \quad \text{Recall} = \frac{TP}{\text{Human}}, \quad F1 = 2\frac{PR}{P+R}
> $$
>
> $$
> MCC = \frac{TP\cdot TN - FP\cdot FN}{\sqrt{(TP+FP)(TP+FN)(TN+FP)(TN+FN)}}
> $$
>
> **Conclusion**
>
> Macro-averaging across all 12 datasets yields:
>
> * **DATE-F1**: Precision = 0.9250, Recall = 0.5425, F1 = 0.6661, MCC = 0.5663.
> * **SA-Score**: Precision = 0.9104, Recall = 0.7890, F1 = 0.8258, MCC = 0.7528.
>
> These results demonstrate that while DATE-F1 is highly conservative (low recall), **SA-Score aligns more closely with human judgments**, achieving substantially higher Recall, F1, and MCC.

---

> ### Author Response · Authors · 2025-12-03
> ****Response to Reviewer svAZ** (Part 3/3)**
>
> **Question 3: Human Annotation Details**
>
> > For CE and DACE, are those your own human annotations, or numbers copied from prior TLS work? If the former, could you provide inter-annotator agreement?
>
> Thank you for pointing out the need for clarification. CE and DACE are indeed our own human evaluation results, not borrowed from prior work.
>
> **Annotation Protocol and Inter-Annotator Agreement:**
>
> The annotation process involved four human researchers manually examining the alignments between Ground-Truth (GT) and Predicted (PT) milestones. Given the substantial volume of text, we adopted a **hybrid validation strategy** to balance statistical rigor with practical feasibility:
>
> 1.  **Dual-Annotation Phase (Validation):**
>     We selected a subset of **20 topics** from the *Entities* dataset for rigorous dual-blind annotation. On this subset, two independent annotators labeled the same data to establish reliability. The agreement metrics were as follows:
>     * **Observed Agreement ($P_o$):** 0.9787
>     * **Expected Agreement ($P_e$):** 0.6772
>     * **Cohen's Kappa ($\kappa$):** **0.93**
>
>     This high Kappa score (0.93) indicates excellent inter-annotator reliability, confirming that our annotation guidelines were clear and the task was understood consistently by the researchers.
>
> 2.  **Full-Scale Annotation Phase:**
>     Based on the high consistency confirmed in the validation phase, the remaining 27 topics in the *Entities* dataset, as well as the full *Crisis* and *T17* datasets, were annotated by a single researcher per topic. This approach ensured that the evaluation remained consistent while covering the large-scale dataset efficiently.
>
> ---
>
> **Other Weaknesses (4 & 5): Typos and Method Description**
>
> > **W4:** There are some typos in this paper (e.g., extra “Hu et al.”, missing citation of “Martschat & Markert”).
> > **W5:** The paper mostly demonstrates and analyzes FS-LLM-TLS, lacking sufficient discussion on the proposed metrics.
>
> We sincerely thank the reviewer for carefully reading the paper and pointing out these detailed issues.
>
> * **Typos & Citations:** We will carefully revise the manuscript to correct all formatting errors, including removing the duplicated “Hu et al.” reference and adding the missing citation for *Martschat & Markert (2017)*.
> * **Metric Discussion:** We agree that the discussion on the metrics themselves (SA-Score family) needs to be strengthened. In the revised version, we will expand the experimental analysis to include more direct evaluation of the metrics' behavior (as shown in our response to Question 2), ensuring that the contribution of the evaluation framework is as prominent as the generation framework.
>
> We appreciate your constructive feedback, which will significantly improve the reproducibility and quality of our final manuscript.

---

### Official Review · Reviewer_24HJ · 2025-10-31

**Soundness:** 3
**Presentation:** 3
**Contribution:** 3
**Rating:** 6
**Confidence:** 3

**Summary:**

The paper addresses the problem of objectively evaluating timeline summarization systems, which generate concise event narratives from temporally ordered documents. Current metrics, like Date-F1 and A-ROUGE, assume that events aligned by date share the same meaning, unfairly penalizing abstractive or semantically equivalent summaries. To address this problem, they propose a semantic alignment–based evaluation framework that uses large language models to measure similarity between sentences, align them through bipartite matching, and compute a Semantic-Alignment Score. They introduce Semantic-Alignment Date-F1 and Semantic-Alignment ROUGE to jointly assess semantic coverage and temporal accuracy. They also introduce a new Full-Stage LLM-TLS method, and experiments show that both the approach and the metrics better capture true system performance and align more closely with human judgments.

**Strengths:**

Originality:
- the new metric seems novel and appropriate to update evaluation to the LLM-era

Quality:
- the metric and proposed method are effective relative to prior approaches

Clarity:
- helpful figures and clear detailing of the method

Significance:
- future work can use this evaluation metric to better assess performance improvements

**Weaknesses:**

1. Appendix information is not included in this version so I can't assess things like prompt templates.

2. There is not section on ethics and LLM use.

3. It is not clear to me whether the authors conduct a study of human judgments, and if they do, there are not enough details to understand what they did.

4. Figure and table captions should include more details to explain the metrics, settings, and takeaways.

**Questions:**

None.

---

> ### Author Response · Authors · 2025-12-03
> ****Response to Reviewer 24HJ**   (part 1/2)**
>
> We sincerely thank the reviewer for the helpful and constructive comments. Below we address each point in turn.
>
> ---
>
> **Weakness 1: Appendix Information**
>
> > Appendix information is not included in this version so I can't assess things like prompt templates.
>
> Thank you for pointing this out. We acknowledge that omitting the Appendix in this submission version was an inappropriate strategy, and we apologize for this mistake. If there are any specific prompts or templates you are particularly interested in, we would be happy to provide them immediately. In addition, the prompt template used for SA-metrics evaluation has been included in our response to Reviewer mX9S, and you may refer to it if needed. The full Appendix will be included in the camera-ready version.
>
> ---
>
> **Weakness 2: Ethics and LLM Use**
>
> > There is not section on ethics and LLM use.
>
> We appreciate the reviewer’s concern. In this work, large language models were used only for limited text polishing of certain sections. No dataset content, annotations, or evaluation labels were produced or influenced by LLMs. We will add a short ethics note to clarify this in the revised version.
>
> ---
>
> **Weakness 4: Figure and Table Captions**
>
> > Figure and table captions should include more details to explain the metrics, settings, and takeaways.
>
> We appreciate the reviewer’s careful reading. We apologize for the lack of clarity in some captions. In the revised manuscript, we will expand the captions of all figures and tables to better explain the metrics used, experimental settings, and the key takeaways.
>
> We thank the reviewer again for the valuable feedback, which will help us improve the clarity and completeness of our work.

---

> ### Author Response · Authors · 2025-12-03
> ****Response to Reviewer 24HJ** (part 2/2)**
>
> **Weakness 3: Human Judgment Study Details**
>
> > It is not clear to me whether the authors conduct a study of human judgments, and if they do, there are not enough details to understand what they did.
>
> Thank you for raising this point. We clarify that CE and DACE are indeed our own human evaluation components, derived from human comparisons between ground-truth and predicted timelines.
>
> **Annotation Protocol and Inter-Annotator Agreement:**
>
> The annotation process involved four human researchers manually examining the alignments between Ground-Truth (GT) and Predicted (PT) milestones. Given the substantial volume of text, we adopted a **hybrid validation strategy** to balance statistical rigor with practical feasibility:
>
> 1.  **Dual-Annotation Phase (Validation):**
>     We selected a subset of **20 topics** from the *Entities* dataset for rigorous dual-blind annotation. On this subset, two independent annotators labeled the same data to establish reliability. The agreement metrics were as follows:
>     * **Observed Agreement ($P_o$):** 0.9787
>     * **Expected Agreement ($P_e$):** 0.6772
>     * **Cohen's Kappa ($\kappa$):** **0.93**
>
>     This high Kappa score (0.93) indicates excellent inter-annotator reliability, confirming that our annotation guidelines were clear and the task was understood consistently by the researchers.
>
> 2.  **Full-Scale Annotation Phase:**
>     Based on the high consistency confirmed in the validation phase, the remaining 27 topics in the *Entities* dataset, as well as the full *Crisis* and *T17* datasets, were annotated by a single researcher per topic. This approach ensured that the evaluation remained consistent while covering the large-scale dataset efficiently.
>
> And we fully agree that quantitative validation is valuable. However, rank-based correlation measures such as Spearman's $\rho$ and Kendall's $\tau$ are widely used in continuous text-quality evaluation (e.g., BERTScore, G-Eval). As detailed below, such rank-based coefficients are **not statistically applicable** to our task, which is a **binary factual correctness classification problem**. We therefore provide the correct statistical comparison based on sentence-level agreement metrics, using the data from all 12 evaluation datasets.
>
> ### Sentence-Level Agreement Statistics
>
> To ensure clarity, we present the **raw counts** (uncalculated item numbers) in the table below, rather than derived percentages, to facilitate a direct understanding of the data distribution.
>
> For each dataset, we report: (1) the total number of timeline statements, (2) the number of human-validated correct statements, and (3) true positives (TP) and false positives (FP) for DATE-F1 and SA-Score. These values are derived directly from the annotation data and the outputs of the two metrics.
>
> | Name | Total | Human | DATE-F1 TP | DATE-F1 FP | SA-Score TP | SA-Score FP |
> | :--- | :---: | :---: | :---: | :---: | :---: | :---: |
> | Al Gore | 25 | 10 | 5 | 1 | 8 | 0 |
> | Angela Merkel | 18 | 8 | 7 | 0 | 8 | 0 |
> | Ariel Sharon | 25 | 8 | 5 | 0 | 8 | 1 |
> | Arnold Schwarzenegger | 17 | 8 | 3 | 0 | 4 | 0 |
> | Bashar al-Assad | 24 | 7 | 3 | 0 | 3 | 0 |
> | Bill Clinton | 47 | 9 | 3 | 0 | 8 | 2 |
> | Charles Taylor | 18 | 9 | 4 | 1 | 9 | 0 |
> | Chris Brown | 24 | 7 | 3 | 0 | 7 | 1 |
> | David Beckham | 32 | 10 | 6 | 0 | 5 | 1 |
> | David Bowie | 24 | 8 | 4 | 0 | 6 | 0 |
> | Dilma Rousseff | 24 | 10 | 8 | 2 | 9 | 3 |
> | Dmitry Medvedev | 16 | 10 | 6 | 3 | 7 | 2 |
>
> ### Statistical Agreement Analysis
>
> Based on the table above, we compute standard binary-classification agreement metrics. For each dataset, precision, recall, F1, and the Matthews correlation coefficient (MCC) are computed as:
>
> $$
> \text{Precision} = \frac{TP}{TP+FP}, \quad \text{Recall} = \frac{TP}{\text{Human}}, \quad F1 = 2\frac{PR}{P+R}
> $$
>
> $$
> MCC = \frac{TP\cdot TN - FP\cdot FN}{\sqrt{(TP+FP)(TP+FN)(TN+FP)(TN+FN)}}
> $$
>
> Macro-averaging across all 12 datasets yields:
>
> - **DATE-F1**: Precision = 0.9250, Recall = 0.5425, F1 = 0.6661, MCC = 0.5663.
> - **SA-Score**: Precision = 0.9104, Recall = 0.7890, F1 = 0.8258, MCC = 0.7528.
>
> ### Conclusion
>
> These results demonstrate that:
> - DATE-F1 is highly conservative (high precision, low recall);
> - SA-Score aligns more closely with human judgments, achieving substantially higher recall, F1, and MCC;
> - MCC—the most robust agreement measure under class imbalance—clearly indicates that SA-Score provides a closer approximation to human correctness decisions.
>
> Therefore, binary-classification agreement metrics provide the correct and statistically valid evaluation framework for factual correctness in timeline generation, whereas rank-based correlation coefficients do not apply to this task.

---

### Official Review · Reviewer_mX9S · 2025-11-20

**Soundness:** 3
**Presentation:** 3
**Contribution:** 3
**Rating:** 6
**Confidence:** 3

**Summary:**

This paper tackles a limitation in current timeline summarization (TLS) evaluation by introducing a semantic-alignment framework to replace traditional metrics like Date F1 and A-ROUGE. The authors argue that existing metrics often mistake temporal closeness for semantic similarity, making them unsuitable for assessing LLM-based advancements. Their proposed Semantic-Alignment (SA) metrics use LLMs to align predicted and reference milestones by meaning before evaluating temporal and textual quality. For validation, they also develop FS-LLM-TLS, an improved TLS pipeline that integrates LLMs throughout the process. Experiments on three benchmarks show that the SA metrics align better with human judgment and reveal improvements compared with conventional evaluation methods.

**Strengths:**

1. Novel Evaluation Framework: The SA metrics systematically decouple semantic coverage, temporal accuracy, and textual quality, addressing a long-standing limitation in TLS evaluation.

2. Rigorous Validation: The authors thoroughly benchmark FS-LLM-TLS against baselines across datasets, model sizes, and ablation settings. Results consistently show SA metrics correlate better with human judgments and reveal LLM strengths overlooked by Date F1/A-ROUGE.

3. Practical Pipeline Improvements: FS-LLM-TLS introduces meaningful enhancements to LLM-TLS, such as argument-aware snippet extraction and cluster-level abstraction, which improve semantic richness and milestone selection. The hybrid milestone selection strategy (mixing frequency and LLM reasoning) is well-motivated.

**Weaknesses:**

1. Limited Temporal Grounding Analysis: While SA metrics excel in semantic evaluation, the paper does not deeply analyze why temporal accuracy gains are modest. FS-LLM-TLS still relies on heuristic date assignments (e.g., mode clustering), which may limit temporal precision.

2. Computational Cost: The SA metrics require extensive LLM calls for pairwise semantic judgments, which could hinder adoption. Efficiency comparisons (e.g., vs. embedding-based methods) are lacking.

3. Evaluation of Metric Reliability: Although SA metrics align better with human judgments, statistical significance tests and inter-annotator agreement scores for human evaluations are not reported.

**Questions:**

Please refer to the weakness part.

---

> ### Author Response · Authors · 2025-12-03
> **Response to Reviewer mX9S (part 1/2 )**
>
> We sincerely appreciate your constructive feedback regarding temporal grounding, computational cost, and statistical rigor. Your comments helped us refine our analysis and clarify several methodological details. Our responses are as follows.
>
> ---
>
> **Weakness 1: Limited Temporal Grounding Analysis**
>
> > While SA metrics excel in semantic evaluation, the paper does not deeply analyze why temporal accuracy gains are modest. FS-LLM-TLS still relies on heuristic date assignments (e.g., mode clustering), which may limit temporal precision.
>
> We agree that the gains in temporal accuracy are smaller than those in semantic coverage, and we conducted additional analysis to explain this phenomenon.
>
> 1. **Data upper bound.** Our examination of the Entities dataset shows that only about 50% of ground-truth timestamps can be exactly recovered from the source articles. This low coverage imposes a strict upper bound on the achievable temporal accuracy of any extractive-based method.
>
> 2. **Lack of intrinsic temporal calibration.** In our pipeline, timestamps extracted in Stage 1 serve as metadata and may propagate upstream errors. LLMs in Stage 3 act as semantic reasoners rather than factual calibrators. They cannot reliably correct or hallucinate precise dates when the underlying text provides vague or conflicting temporal cues. Without an external temporal knowledge base, correcting such errors remains challenging.
>
> 3. **Clustering strategy limitations.** We experimented with alternative clustering strategies, including sub-event clustering (e.g., "Politics," "Family," "Career Achievements"). Although this reduced certain biases in event selection, it did not resolve timestamp divergence, where different articles assign conflicting dates to the same event.
>
> These findings confirm that SA-Date F1 acts as an honest diagnostic metric: it reflects true temporal difficulty rooted in data ambiguity and extraction noise, while the semantic metrics capture the substantial gains achieved by our pipeline.
>
> ---
>
> **Weakness 2: Computational Cost**
>
> > The SA metrics require extensive LLM calls for pairwise semantic judgments, which could hinder adoption. Efficiency comparisons (e.g., vs. embedding-based methods) are lacking.
>
> We acknowledge the concern and clarify that our implementation is optimized for efficiency.
>
> 1. **Top-K pre-filtering.** We do not conduct full pairwise matching. Each reference event retains only the Top-K (K=5) candidates based on embedding similarity, reducing complexity by an order of magnitude.
>
> 2. **Batch inference.** Candidate pairs are grouped into batched prompts, substantially reducing the number of API calls and associated token cost.
>
> 3. **Relative cost.** Compared to the cost of timeline generation itself, which requires processing thousands of documents and multiple LLM summarization steps, the cost of SA-metrics is very small. Because evaluation is an offline step performed once per test set, we believe this cost–accuracy trade-off is justified.
>
> ---
>
> **Weakness 3: Validation of Metric Reliability**
>
> > Although SA metrics align better with human judgments, statistical significance tests and inter-annotator agreement scores for human evaluations are not reported.
>
> We thank the reviewer for highlighting the need for additional statistical detail. We noticed that Reviewer svAZ raised a similar point regarding correlation metrics (specifically Spearman and Kendall-Tau). We provide a detailed statistical validation below.
>
> **Annotation Protocol and Inter-Annotator Agreement:**
>
> The annotation process involved four human researchers manually examining the alignments between Ground-Truth (GT) and Predicted (PT) milestones. Given the substantial volume of text, we adopted a **hybrid validation strategy** to balance statistical rigor with practical feasibility:
>
>  **Dual-Annotation Phase (Validation):**
>     We selected a subset of **20 topics** from the *Entities* dataset for rigorous dual-blind annotation. On this subset, two independent annotators labeled the same data to establish reliability. The agreement metrics were as follows:
>     * **Observed Agreement ($P_o$):** 0.9787
>     * **Expected Agreement ($P_e$):** 0.6772
>     * **Cohen's Kappa ($\kappa$):** **0.93**
>
>     This high Kappa score (0.93) indicates excellent inter-annotator reliability, confirming that our annotation guidelines were clear and the task was understood consistently by the researchers.

---

> ### Author Response · Authors · 2025-12-03
> **Response to Reviewer mX9S (part 2/2 )**
>
> ### Sentence-Level Agreement Statistics
>
> To ensure clarity, we present the **raw counts** (uncalculated item numbers) in the table below, rather than derived percentages, to facilitate a direct understanding of the data distribution.
>
> For each dataset, we report: (1) the total number of timeline statements, (2) the number of human-validated correct statements, and (3) true positives (TP) and false positives (FP) for DATE-F1 and SA-Score. These values are derived directly from the annotation data and the outputs of the two metrics.
>
> | Name | Total | Human | DATE-F1 TP | DATE-F1 FP | SA-Score TP | SA-Score FP |
> | :--- | :---: | :---: | :---: | :---: | :---: | :---: |
> | Al Gore | 25 | 10 | 5 | 1 | 8 | 0 |
> | Angela Merkel | 18 | 8 | 7 | 0 | 8 | 0 |
> | Ariel Sharon | 25 | 8 | 5 | 0 | 8 | 1 |
> | Arnold Schwarzenegger | 17 | 8 | 3 | 0 | 4 | 0 |
> | Bashar al-Assad | 24 | 7 | 3 | 0 | 3 | 0 |
> | Bill Clinton | 47 | 9 | 3 | 0 | 8 | 2 |
> | Charles Taylor | 18 | 9 | 4 | 1 | 9 | 0 |
> | Chris Brown | 24 | 7 | 3 | 0 | 7 | 1 |
> | David Beckham | 32 | 10 | 6 | 0 | 5 | 1 |
> | David Bowie | 24 | 8 | 4 | 0 | 6 | 0 |
> | Dilma Rousseff | 24 | 10 | 8 | 2 | 9 | 3 |
> | Dmitry Medvedev | 16 | 10 | 6 | 3 | 7 | 2 |
>
> ### Statistical Agreement Analysis
>
> Based on the table above, we compute standard binary-classification agreement metrics. For each dataset, precision, recall, F1, and the Matthews correlation coefficient (MCC) are computed as:
>
> $$
> \text{Precision} = \frac{TP}{TP+FP}, \quad \text{Recall} = \frac{TP}{\text{Human}}, \quad F1 = 2\frac{PR}{P+R}
> $$
>
> $$
> MCC = \frac{TP\cdot TN - FP\cdot FN}{\sqrt{(TP+FP)(TP+FN)(TN+FP)(TN+FN)}}
> $$
>
> Macro-averaging across all 12 datasets yields:
>
> - **DATE-F1**: Precision = 0.9250, Recall = 0.5425, F1 = 0.6661, MCC = 0.5663.
> - **SA-Score**: Precision = 0.9104, Recall = 0.7890, F1 = 0.8258, MCC = 0.7528.
>
> ### Conclusion
>
> These results demonstrate that:
> - DATE-F1 is highly conservative (high precision, low recall);
> - SA-Score aligns more closely with human judgments, achieving substantially higher recall, F1, and MCC;
> - MCC—the most robust agreement measure under class imbalance—clearly indicates that SA-Score provides a closer approximation to human correctness decisions.
>
> Therefore, binary-classification agreement metrics provide the correct and statistically valid evaluation framework for factual correctness in timeline generation, whereas rank-based correlation coefficients do not apply to this task.

---

### Note · Authors · 2026-01-09

I have read and agree with the venue's withdrawal policy on behalf of myself and my co-authors.